# Proteomics Approach of Rapamycin Anti-Tumoral Effect on Primary and Metastatic Canine Mammary Tumor Cells In Vitro

**DOI:** 10.3390/molecules26051213

**Published:** 2021-02-25

**Authors:** Patrícia F. Lainetti, Antonio F. Leis-Filho, Priscila E. Kobayashi, Laíza S. de Camargo, Renee Laufer-Amorim, Carlos E. Fonseca-Alves, Fabiana F. Souza

**Affiliations:** 1Department of Veterinary Surgery and Animal Reproduction, School of Veterinary Medicine and Animal Science, São Paulo State University—UNESP, Botucatu 18618-681, Brazil; patricia.lainetti@unesp.br (P.F.L.); laizacamar@gmail.com (L.S.d.C.); carlos.e.alves@unesp.br (C.E.F.-A.); 2Department of Veterinary Clinic, School of Veterinary Medicine and Animal Science, São Paulo State University—UNESP, Botucatu 18618-681, Brazil; nandoleis@hotmail.com (A.F.L.-F.); priscila.e.kobayashi@unesp.br (P.E.K.); renee.laufer-amorim@unesp.br (R.L.-A.); 3Institute of Health Sciences, Universidade Paulista—UNIP, Bauru 17048-290, Brazil

**Keywords:** cell culture, female dog, neoplasia, protein

## Abstract

Rapamycin is an antifungal drug with antitumor activity and acts inhibiting the mTOR complex. Due to drug antitumor potential, the aim of this study was to evaluate its effect on a preclinical model of primary mammary gland tumors and their metastases from female dogs. Four cell lines from our cell bank, two from primary canine mammary tumors (UNESP-CM1, UNESP-CM60) and two metastases (UNESP-MM1, and UNESP-MM4) were cultured in vitro and investigated for rapamycin IC_50_. Then, cell lines were treated with rapamycin IC_50_ dose and mRNA and protein were extracted in treated and non-treated cells to perform AKT, mTOR, PTEN and 4EBP1 gene expression and global proteomics by mass spectrometry. MTT assay demonstrated rapamycin IC_50_ dose for all different tumor cells between 2 and 10 μM. RT-qPCR from cultured cells, control versus treated group and primary tumor cells versus metastatic tumor cells, did not shown statistical differences. In proteomics were found 273 proteins in all groups, and after data normalization 49 and 92 proteins were used for statistical analysis for comparisons between control versus rapamycin treatment groups, and metastasis versus primary tumor versus metastasis rapamycin versus primary tumor rapamycin, respectively. Considering the two statistical analysis, four proteins, phosphoglycerate mutase, malate dehydrogenase, l-lactate dehydrogenase and nucleolin were found in decreased abundance in the rapamycin group and they are related with cellular metabolic processes and enhanced tumor malignant behavior. Two proteins, dihydrolipoamide dehydrogenase and superoxide dismutase, also related with metabolic processes, were found in higher abundance in rapamycin group and are associated with apoptosis. The results suggested that rapamycin was able to inhibit cell growth of mammary gland tumor and metastatic tumors cells in vitro, however, concentrations needed to reach the IC_50_ were higher when compared to other studies.

## 1. Introduction

Mammary gland tumors are frequent in female dogs (*Canis lupus familiaris*), as well as in women. They represent about 50% of the tumors diagnosed in dogs and the majority are malignant [1]. Carcinoma is the most common malignant subtype of these tumors [2]. Metastases in lymph nodes and/or distant organs (lungs and bones) occur frequently, since 30% of dogs with malignant mammary gland tumors have metastases [3]. The main treatment for mammary tumors in dogs is mastectomy. However, in more aggressive tumors or in cases of recurrence and metastases, chemotherapy as a complementary treatment to surgery or even as a palliative treatment may be used [4]. In the case of female dog, chemotherapy for mammary tumors does not have effective protocols and the protocols used include doxorubicin associated with cyclophosphamide, cisplatin, carboplatin and gemcitabine. These drugs usually are chosen according to previously results obtained in human studies [4,5].

Cell culture of neoplastic cells have been used as a preclinical experimental model in tests with tumor cells in humans, being considered an alternative to in vivo models [6]. This model allows carrying out pharmacological cell toxicity tests that assist in the discovery of new therapies and in the identification of the pathways in which the drugs tested act [7,8]. Rapamycin is an antifungal, with immunosuppressive and antiproliferative activity, tested in vitro as a chemotherapy agent to be used in women’s breast cancer [9,10] and in vivo in rats [11]. This drug acts in the AKT-mTOR pathway in different species, including in canine osteosarcoma [12,13].

The AKT-mTOR pathway is closely linked to cell proliferation and growth, protein synthesis and apoptosis. This pathway is an important modulator of the aging process and diseases related to it. In the presence of neoplasms, this pathway is incorrectly regulated, as there is an increase in the activation of the TOR complex (mTOR) by mutations of function gain in oncogenes and loss of tumor suppressor genes [14]. The inhibition of mTOR complex can be a strategy for the development of new therapies against neoplasms since it would inhibit cell proliferation and induce apoptosis [15]. 

Therefore, the aim of this study was to evaluate the effects of rapamycin treatment on viability, gene expression and proteomics of canine mammary tumor cells cultured in vitro, from two primary tumors and their metastases, since this drug was not yet studied as a chemotherapy agent to treat canine mammary tumor cells.

## 2. Results

### 2.1. Mammary Carcinoma Tissue

Primary tumors and their metastases, in immunofluorescence, showed positive expression for AKT, mTOR, PTEN and 4EBP1 proteins, as seen in Appendix A. In addition, primary tumors (*n* = 2) and metastases (*n* = 2) mean ± standard error of protein expression are simplified in Appendix A.

### 2.2. Cell Culture

#### 2.2.1. Cellular Metabolic Activity Evaluation

MTT assay was performed in all cell lines and it showed decreased cellular metabolic activity in all concentrations and periods of exposure to the drug. UNESP-CM1 and UNESP-MM1 had approximately IC_50_ of 10 and 4 µM at 24 h, respectively. UNESP-CM60 and UNESP-MM4 had approximately IC_50_ of 9 and 6 µM at 24 h, respectively. Different doses and responses percentage, in addition to different periods of exposure can be seen in Appendix A.

UNESP-CM1 and UNESP-CM60, primary tumors, showed more resistance to the drug and cell viability decreased in highest concentrations of rapamycin at 24 h. This decreased was not as abrupt at 48 and 72 h, except at the concentration of 12 µM in UNESP-CM60 cells. In the case of UNESP-MM1 and UNESP-MM4, cell viability was similar in all doses and treatment periods. In general, a lower viability was observed at 48 and 72 h, mainly in the 10 and 12 µM treatments in UNESP-MM4. Cellular behavior in all treatments, both in primary tumors and in their metastases, were different. In general, the majority of cell viability was reduced overtime.

#### 2.2.2. RT-qPCR

RT-qPCR transcripts were analyzed comparing control and rapamycin treatment groups and then, comparing transcripts expression between primary and metastatic cells. There was no statistical difference in transcripts expression analyzed in the two different groups (control versus treated groups and primary versus metastatic cells), as demonstrated in Appendix A.

#### 2.2.3. Proteomics

A total of the 273 proteins were found in all groups. Considering control (metastasis and primary tumor) and rapamycin treatment (metastasis and primary tumor) 111 and 116 proteins, were respectively observed (Figure 1A). In the comparisons between metastasis control, primary tumor control, metastasis rapamycin and primary tumor rapamycin 162, 168, 167 and 174 proteins were respectively found (Figure 1B). Proteins found in the different groups are listed by name, ID, gene and gene ontology (molecular function, biological process and cellular component), as can be observed in Appendix A. Mass spectrometry data from all proteins are in Appendix A.

For control versus rapamycin group, the results of gene ontology were similar (Figure 2). In addition, primary tumor control versus metastases control versus primary tumor rapamycin versus metastases rapamycin, gene ontology presented a regular pattern (Figure 3). The main molecular function, biological process and cellular component were catalytic activity, cellular process, and cell, respectively.

PLS-DA represented by VIP score indicated 13 proteins, considering α > 1.6 for control versus rapamycin groups (Figure 4). Similar proteins were also significant in the test t and the Volcano plot analysis (FDR < 0.05).

Considering metastasis control, primary tumor control, metastasis rapamycin and primary tumor rapamycin groups the PLS-DA represented by VIP score indicated 22 proteins with α > 1.6 (Figure 5). However, 4 proteins (phosphoglycerate mutase [E2RT65], dihydrolipoyl dehydrogenase [F1PAR0], uncharacterized protein [E2RKQ6] and malate dehydrogenase [F1PYG8, Q0QF34]) were also significant in the ANOVA one-way analysis (FDR < 0.05). For uncharacterized protein (E2RKQ6) and malate dehydrogenase (F1PYG8, Q0QF34) were observed differences between all groups, but for phosphoglycerate mutase (E2RT65) and dihydrolipoyl dehydrogenase (F1PAR0) differences were observed between primary tumor treated with rapamycin and all other groups. There was no difference between the other groups to the last two proteins.

## 3. Discussion

Cells used for the experiment were characterized as HER-2 overexpressing or triple negative basal like subtypes, which are associated with higher chances of tumor recurrence and worst prognosis, respectively [16,17,18,19]. These two different molecular subtypes were chosen to the experiment because of their importance due to their aggressive behavior, higher chances of developing metastasis [17,20]. In addition, those subtypes can be considered an important model to be used in women breast cancer research, since their behavior and characteristics are similar in both species [17,19]. HER-2 overexpressing tumors are not well comprehended in veterinary oncology but showed more cell pleomorphism and higher number of mitosis figures [16,21] as observed in humans [22]. Studying this type of tumor could help to understand their importance and to choose an appropriate treatment approach in dogs. Triple-negative basal like tumors in dogs are as aggressive as in humans and have an unfavorable prognosis, because do not express receptors for HER-2, estrogen (ER) and progesterone (PR) [18,20].

Immunofluorescence of the mammary gland tissue was used to check the expression of the selected proteins (AKT, mTOR, PTEN and 4EBP1). In addition, this result validates the RT-qPCR performed on cells cultured in vitro, proving that the results obtained from cell culture are reliable and can be used as an experimental model in vitro, in this species. Such observations have already been mentioned in humans [7,8]. The proposed model in this article might be used as an experimental model for humans as well, since tumor characteristics such as age at tumors development, hormonal influence, genes expressed in oncogenesis and the proteins produced have similarities between those species [18,23]. However, cells can undergo genetic changes during growth in vitro [24], which can interfere with the evaluations performed. Protein expression and gene expression of AKT, mTOR, PTEN and 4EBP1 in tissue samples, primary tumors cells and their respective metastasis demonstrates that these changes were not acquired or changed during the cell expansion process.

MTT assay showed that the tumor cells responded to the treatment with rapamycin, identifying the approximately IC_50_ in concentrations of 9, 6, 10 and 4 µM, with 24 h of treatment, for UNESP-CM60, UNESP-MM4, UNESP-CM1 and UNESP-MM1, respectively. Exposure to the drug for more than 24 h impaired cellular metabolic activity, in the same way that it inhibited the growth of breast tumor cells in women and rats [9,11]. Rapamycin concentrations varied in the different tumor types studied. Apparently, each cell culture had an IC_50,_ even with all of them expressing AKT, mTOR, PTEN and 4EBP1 proteins.

Different IC_50_ among the cell cultures indicate that some cell types have a higher sensitivity to rapamycin action. As in rats, in which the activity of rapamycin has been shown to be dose-dependent, both in vivo and in vitro [11,25]. Rapamycin concentrations used to obtain the IC_50_ were similar in UNESP-CM60 and UNESP-CM1. In UNESP-MM4 and UNESP-MM1 cells however, the concentrations used were different. UNESP-MM4 cells were more sensitive to rapamycin in 24 h and required a lower concentration of the drug to achieve cell inhibition, when compared to UNESP-CM60, and this also occurred with UNESP-MM1 and UNESP-CM1. Moreover, cells HER-2 overexpressing (UNESP-CM1, UNESP-CM60 and UNESP-MM4) had the higher doses to achieve IC_50_ than triple negative non-basal tumor cells (UNESP-MM1), which indicated that rapamycin could have a better response on triple negative cells when compared to HER-2 overexpressing cells. This can occur due to the different ways of activation of PI3K pathway by HER-2, HER-3, HER-4 and EGFR activated receptors in HER2 overexpressing tumors [26] and because PI3KCA gene is most frequent mutated gene in triple negative breast cancer, just after TP53 [27].

Molecular functions associated with proteins found were glycolytic process, cell redox homeostasis, mitotic cell cycle, carbohydrate metabolic process, regulation of RNA metabolic process, mitotic cell cycle and reactive oxygen species metabolic process. In proteomics, considering the two statistical analysis, phosphoglycerate mutase, l-lactate dehydrogenase, malate dehydrogenase, 14-3-3 binding protein and nucleolin were found in higher abundance in control group. Moreover, NAD(P)H quinone dehydrogenase, superoxide dismutase, dihydrolipoyl dehydrogenase and carboxypeptidase were increased abundance in the cell group treated with rapamycin.

Phosphoglycerate mutase (PGAM1) is a glycolytic enzyme that is involved in glycolysis and metabolic activity in tumor growth, survival, and invasion [28,29]. It catalyzes the conversion of 3-phosphoglycerate (3-PG) to 2-phosphoglycerate (2-PG) and thus initiates aerobic glycolysis. This protein, when overexpressed, is involved in cancer progression and has already been found in breast, lung, and prostate tumors, acting on the proliferation and metabolism of cancer cells. Inactivation of PGAM1 gene increases apoptosis rates and decreases tumor proliferation and growth [28,29,30,31]. In prostate tumors, its inactivation led to the inhibition of cell proliferation, migration and invasion and increased the apoptosis of tumor cells [31], and in breast tumors it was related to tumor cell migration [28,29]. Moreover, its blockage can be used as a therapeutic target for tumors that express this protein [28,32]. A relationship has already been established between the drug Bevacizumab and decreased expression of the PGAM1 gene [33]. However, the expression of PGAM1 has been found to be decreased in cells resistant to methotrexate, which can change glucose metabolism and be involved in the mechanism of multiple drug resistance in breast tumors [34]. PGAM1 was found with decreased expression in primary and metastatic mammary tumors cells that were treated with rapamycin, which may show a possible rapamycin suppression on this protein. PGAM1 was found in higher abundance in metastatic mammary tumor cells than primary tumor cells, which coincides with its tumor progression function, which is observed in higher amounts.

Malate dehydrogenase (MDH) is known to be one of the sources of nicotinamide adenine dinucleotide (NAD^+^), allowing glycolysis cycle continuation [35]. The enzyme is important for the metabolism of malignant tumors and has already been found in different tissues, such as breast, prostate and non-small cell lung carcinoma tumors [36,37,38]. Increased MDH activity can aid in tumorigenesis and proliferation of tumor cells in stressful tumor environments [35], is associated with higher invasion capacity [37], and has already been linked to tumor resistance of prostatic cells to docetaxel [36]. In the case of mammary tumor cells from female dogs grown in vitro, MDH expression was increased in primary tumors when compared to metastatic tumors, which may be related to the tumor progression of primary tumors until they become metastases. In addition, MDH was decreased in cells treated with rapamycin, when compared to untreated ones, which corroborates the finding by Naik et al. [39], which showed that when the expression of MDH is suppressed, there is a decrease in tumor growth.

L-lactate dehydrogenase (LDH) is involved in pyruvate to lactate conversion, generating NAD^+^ and participating in the glycolysis pathway and it can be divided into 2 subunits, LDHA and LDHB [40,41]. The production of different NAD^+^ precursors (such as MDH and LDH), helps in cancer cells proliferation [41,42]. LDHB expression does not have a pattern like LDHA, it can vary depending on the tumor type and the tumor’s dependence on aerobic glycolysis [41,43]. However, in triple negative/basal-like breast tumors, LDHB is increased [43]. Its high expression is also related as a predictor of lower survival rates. The mTOR pathway is a positive regulator of LDH [43,44], this may be one of the reasons why LDH was found with less expression in cells treated with rapamycin, when compared to control group, since rapamycin acts inhibiting the AKT-mTOR pathway. In addition, the tumors used in this research were classified as basal-like tumors, which would explain the high expression of LDHB in the control cells.

The 14-3-3 proteins are part of the phosphoserine/phosphothreonine-binding proteins family and are involved in suppressing cell apoptosis, controlling cell cycle and proliferation. They can bind to several other proteins that will define their action [45,46]. The 14-3-3 protein has ability to bind to oncogenic proteins and therefore its overexpression is highly related to several tumor types, such as breast, prostate, ovaries and lung cancer [47,48]. 14-3-3γ, found in this study, is one of the isoforms already identified in 14-3-3 protein, and is related to cell motility [48,49]. Consequently, it is causally related with migration and invasion of tumor cells and the presence of metastases. Similarly, to our study, this isoform has already been linked to metastases of breast tumors and its inhibition has reduced the invasiveness of malignant cells [49]. In any case, regardless of the isoform, inhibition of 14-3-3 protein is strongly linked to inhibition of cell proliferation and migration, in addition to stimulating apoptosis [46,47,48].

Nucleolin overexpression was related with tumorigenesis and tumor progression, since it promotes invasion and angiogenesis, inducing cell growth, proliferation and increasing survival [50,51]. Therefore, nucleolin overexpression is also correlated with a higher risk of tumor recurrence, decreased survival rate to different types of cancer and higher rates of metastasis [51,52]. This protein was found increased in control group, which consisted of mammary tumor cells and their metastases, corroborating with the findings in women [51]. In addition, nucleolin can be a therapeutic target for breast tumors [52] and we showed that the rapamycin was effective in decreasing the expression of this protein, since in the treated group of canine mammary tumor cells cultured in vitro the protein was found in lower abundance.

Annexin A2 is related to angiogenesis and metastasis in breast cancer. Its downregulation is associated with increased apoptosis and decreased cell viability and migration [53]. This protein is related to progression and metastasis in breast tumors, by activating the PI3k/AKT pathway and can be considered a therapeutic target for more aggressive tumors. When inhibited decreases cellular invasion capacity and as a result decreases metastasis risks [54,55]. Annexin A2 protein was found in increased abundance in the control group, of primary and metastatic cells. This result was compatible with the results of the other study, which demonstrated a higher expression of annexin A2 gene in cells with a more malignant phenotype and with negative estrogen receptor [55]. After treatment with rapamycin lower abundance was observed, suggesting an action in the PI3k/AKT pathway inhibiting the expression of this protein.

Dihydrolipoyl dehydrogenase or dihydrolipoamide dehydrogenase (DLD) is a subunit of pyruvate dehydrogenase complex (PDC) which is known to modulate cellular energy metabolism and homeostasis [56,57]. DLD produces reactive oxygen species (ROS) from its oxidoreductive activity [58,59,60]. Normal cells produce ROS as a way of regulating homeostasis, but tumor cells produce even more, as they can withstand higher levels of ROS. However, after exceeding the limit that these cells can withstand, they become more susceptible to ROS action, leading tumor cells to apoptosis. Therefore, the use of ROS may be favorable in the treatment of cancer [59,60]. The use of DLDH, in conjunction with chemotherapy drugs, induces to apoptosis of melanoma cells cultured in vitro, but not of normal cells [59]. DLD protein was found to be increased in both primary tumors and metastases treated with rapamycin. This fact shows that the increase in DLD in cells treated with rapamycin induced increased ROS levels and apoptosis.

NAD(P)H quinone dehydrogenase (NQQ1) acts by regulating the stable expression of the p53 gene and its inhibition is related with the decrease in p53 expression, increasing the susceptibility of tumors to development [61]. Cells treated with rapamycin showed higher abundance of this protein, making it possible to relate the treatment to a stimulation of NQQ1 and consequently an increase in the expression of the tumor suppressor gene p53, a similar result to that found by Paek et al. [62] in women breast tumor cells.

Succinate dehydrogenase (SDH) is a mitochondrial enzyme that participates in both the citric acid cycle and the electron transport chain [63]. SDH is well known as a tumor suppressor [64] and its low expression leads to proliferation and tumorigenesis [63]. Present or overexpressed, it acts as a flag for apoptosis [65]. Rapamycin-treated cells showed high SDH expression when compared to the control group, which may be related to cell proliferation in the control group and apoptosis in the treated group.

The dose used to achieve these results may be considered higher than the dose acceptable to use in animals. The safety rapamycin dose to use in dogs is 0.08 mg/kg [66], but higher doses were not tested. Thus, the in vitro rapamycin dose necessary to inhibit tumor cells may exceed the maximum recommended dose in vivo. 

Some limitations can be mentioned in this study. The first-generation rapamycin (sirolimus) was used, which may have impaired the cellular response to the drug. Currently, new generation drugs have been shown to be more efficient in the treatment of tumors and capable of inhibiting tumor growth at lower doses than are necessary using sirolimus. This limitation of the performance of first-generation rapamycin (sirolimus) in mTORC1 and the absence of action in the mTORC2 unit may explain a resistance to rapamycin and a different cellular sensitivity to this drug [67,68,69]. The mTOR complex is composed of two units, mTORC1 and mTORC2. mTORC1 is known to induce cell proliferation and mRNA translation, and mTORC2 may be related to tumor evolution [69]. There are three generations of mTOR inhibitor compounds, the first generation has a limited action and only on the mTORC1, while the third generation compound have an inhibitory activity of both units [67,68]. In addition, the technique used to perform proteomics has limitations, as there are more recent methods with greater sensitivity. 

Even with similar protein expressions, the response to rapamycin was different in all 4 cell cultures. HER-2 cells needed a higher dose when compared to triple negative cells and primary tumors were more resistant to rapamycin treatment than metastasis cells. This shows an increasingly necessary personalized medicine within veterinary medicine. Our study showed that rapamycin has an anti-tumor action in canine mammary tumor cells cultured in vitro and rapamycin treatment led to changes in important proteins expression, but more studies are needed to understand rapamycin activity associated with other chemotherapeutic drugs and its action as an adjunctive drug.

## 4. Materials and Methods

### 4.1. Ethics Committee

The study was carried out according to the National and International Recommendations for the Care and Use of Animals [70]. All procedures were performed after receiving approval from the Institutional Ethics Committee on Animal Use (#0189/2018).

### 4.2. Experimental Design

Four tumors previously characterized [71] were used to perform immunofluorescence for AKT, PTEN, mTOR and 4EBP1 antibodies. Then, tumor cells were expanded and analyzed for cell viability (MTT), gene expression (RT-qPCR) and proteomics as can be observed in Figure 6.

### 4.3. Mammary Carcinoma Samples, Histological and Immunohistochemical Analysis

Histopathological classification of the tumors was performed according to Zappulli et al. [72] and immunohistochemistry for tumor immunophenotype according to Nguyen et al. [19].

Two mammary carcinomas were obtained from two intact dogs, over 10 years old, and both animals presented metastases in different locations. One patient had bone (humerus) and lung metastases and the other had an inguinal lymph node metastasis. Both animals were classified as stage V in the TNM classification. Histologically, the tumors were classified as a grade II solid carcinoma and an adenosquamous carcinoma. In immunohistochemistry, primary tumors were both classified as HER-2 overexpressing. Grade II solid carcinoma bone metastasis was classified as triple negative with non-basal immunophenotype and adenosquamous carcinoma lymph node metastasis as HER-2 overexpressing, previously described by our group in Lainetti et al. [71].

### 4.4. Immunofluorescence of Tumor Tissue

Paraffin embedded samples of primary and metastatic mammary carcinoma were cut (3 μm), transferred to positive charge slides (StarFrost, Braunschweig, Germany) and deparaffinized. For PTEN, 4EBP1, AKT and mTOR antibodies, the slides were subjected to antigenic retriever with citrate buffer (pH 6.0) in a pressure cooker (Pascal, Dako, Agilent Technologies, Santa Clara, CA, USA). Blocking of endogenous peroxidase was performed with 8% hydrogen peroxide in methanol, for 20 min and subsequently, blocking of non-specific bonds was performed with 8% skimmed milk, for 60 min, both at room temperature. Antibodies were diluted and samples were incubated according to the manufacturer’s guidelines and the information in Appendix A. Sections were stained with primary antibody against p-mTOR (rabbit monoclonal), PTEN (rabbit polyclonal), p-4EBP1 (rabbit monoclonal) and pAKT (rabbit polyclonal) according to previous results by [73,74,75,76]. The detection of antibodies was performed using Alexa Fluor 488 secondary antibody (Life Technologies, Corporation, Carlsbad, CA, USA). Counter staining was performed with DAPI (Sigma Aldrich, St. Louis, MO, USA). As a negative control of the reaction, primary antibodies tested during the procedure were omitted which were replaced with a TRIS buffer solution.

Tissue samples were evaluated using a laser scanning confocal microscope (TCS SP 5, Leica, Weltzlar, Germany). Images were digitized and evaluated by specific software (LAS AF Leica Application Suite Advanced Fluorescence, LAS AF, Weltzlar, Germany). Positive control was normal canine tissues as follow, for mTOR prostate, for 4EBP1 was normal stomach tissue, PTEN was normal prostate and AKT was normal colon, following the information from Protein Atlas (www.proteinatlas.org). Images were digitized (Leica Application Suite Advanced Fluorescence, LAS AF, Weltzlar, Germany) and evaluated by specific software ImageJ (National Institutes of Health, Bethesda, MD, USA). 

### 4.5. Cell Culture

Cultured cells were acquired from our cell bank and previously characterized [68]. Cell culture was developed from tumor fragments in MEGM™ (Lonza Inc., Allendale, NJ, USA) with 1% antibiotic/antimycotic solution (ThermoFisher Scientific, Waltham, MA, USA) and 10% fetal bovine serum (FBS) (LGC Biotecnologia, Cotia, SP, Brazil), kept in a humid atmosphere containing 5% CO_2_ at 37 °C. After the initial cell expansion (minimum 72 h), also all samples were cryopreserved in different passages in MEGM™ (Lonza Inc., Allendale, NJ, USA) with 11.8% DMSO (Sigma Aldrich, St. Louis, MO, USA) and 47% FBS. Remaining samples were expanded and used for cellular metabolic activity evaluation (MTT), gene expression (RT-qPCR) [24] and proteomics.

Cells were obtained from our cell bank from grade II solid carcinoma (UNESP-CM1) and its respective bone metastasis (UNESP-MM1) and adenosquamous carcinoma (UNESP-CM60) and its respective lymph node metastasis (UNESP-MM4).

### 4.6. Rapamycin Treatment and Assessment of Cellular Metabolic Activity

Metabolic activity of cultured primary and metastatic mammary gland tumor cells, treated or not with rapamycin, was determined by the conversion of 3-(4,5-dimethylthiazol-2-yl)-2,5-diphenyltetrazolium bromide (MTT) (Life Technologies Corporation, Carlsbad, CA, USA) which is yellow in formazan (violet). This conversion indicates the number of viable cells. The shade of violet is directly proportional to the concentration of viable cells in the culture plate. 

Tumor cells were transferred to 96-well plates (10.000 cells/well) and after 24 h of incubation in MEGM™ culture medium, rapamycin (diluted in DMSO) was added in 4 concentrations (9, 10, 11 and 12 µM). Rapamycin concentrations were chosen based on previous studies in breast tumor cells [9,10]. After 24, 48 and 72 h, MTT (0.5 mg/mL in DPBS) was added and incubated at 37 °C for 4 h and then, MTT salt were dissolved with DMSO. Analysis of the results was carried out 10 min after the addition of DMSO, in a microplate reader (Biochrom Asys Expert Plus Microplate Reader, Biochrom Ltd, Harvard Bioscience, Holliston, MA, USA), at 550 nm absorbance. MTT was carried out in six repetitions for each rapamycin concentration. A control with DMSO, at the same concentration as the highest dose of rapamycin, was used to ensure that there was no effect of the product on cell proliferation.

Based on test results, maximum inhibitory concentration (IC_50_) of cell viability was determined using the formula: Cell viability % = (absorbance of the treated sample−blank absorbance)/(absorbance control cells DMSO—blank absorbance) × 100; whereas the blank was DPBS and cells in the control group did not came in contact with rapamycin. 

### 4.7. RT-qPCR

For mRNA extraction, cryopreserved cells were thawed in a humid bath at 37 °C, centrifuged (450× *g*, 5 min) and resuspended in MEGM™ with 1% antibiotic/antimycotic solution (ThermoFisher Scientific, Waltham, MA, USA) and 10% FBS. Cells were transferred to 6-well plates and cultured until reaching 70% confluence. Then, they were washed 3 times with DPBS in an ice bath. mRNA extraction followed the protocol recommended by the manufacturer (mini–Kit RNeasy, Qiagen, Hilden, Germany). mRNA concentration was evaluated in a spectrophotometer (NanoDrop™, ND-8000, ThermoFisher Scientific, Waltham, MA, USA). Total RNA was extracted and treated to eliminate any contamination with genomic DNA, with 1 U of DNase I Amplification Grade (Life Technologies, Corporation, Carlsbad, CA, USA) in 10X DNase I Reaction Buffer and 25 mM of pH 8.0 EDTA. Replication of the cDNA was carried out in a thermocycler (Veriti® 96-Well Fast Thermal Cycler, Applied Biosystems, Life Technologies, Corporation, Carlsbad, CA, USA). Reverse transcription was performed according to Rivera-Calderon et al. [74].

Amplifications were evaluated on an automatic thermal cycler (QuantStudio™ 12K Flex Real-Time PCR System, 4471087, Applied Biosystems™, ThermoFisher Scientific, Carlsbad, CA, USA) and processed by the detection system after a variable number of cycles in an exponential phase. Values obtained for all samples were normalized by the ratio obtained between the gene of interest and the reference gene. Endogenous reference genes used were RPS5, RPS19 and HPRT (selected in previous studies of the group) and the three selected genes are described in Appendix A. Primers used for amplification of the genes of interest were designed using the Primer Express 2.0 program (ThermoFisher Scientific, Waltham, MA, USA). For amplification, the Power SYBR Green PCR Master MIX kit (ThermoFisher Scientific, Waltham, MA, USA) was used following the manufacturer’s instructions. Genes selected and validated by RT-qPCR were AKT, PTEN and mTOR and the values to RT-qPCR were represented by relative gene quantification (RQ), calculated by the 2^−ΔΔCT^ method [77].

The experiment was performed with six replicates for each cell. UNESP-CM1, UNESP-CM-60, UNESP-MM1 and UNESP-MM4 were divided in control (not treated cells) and rapamycin treatment group (treatment for 24h). For evaluation, cells were divided in control (UNESP-CM1, UNESP-CM60, UNESP-MM1 and UNESP-MM4) versus rapamycin treatment (UNESP-CM1, UNESP-CM60, UNESP-MM1 and UNESP-MM4), and also in primary tumor cells (UNESP-CM1 and UNESP-CM60) versus metastasis cells (UNESP-MM1 and UNESP-MM4). Cells were also evaluated among themselves, UNESP-CM1 control versus UNESP-CM1 rapamycin treatment; UNESP-CM60 control versus UNESP-CM60 rapamycin treatment; UNESP-MM1 control versus UNESP-MM1 rapamycin treatment and UNESP-MM4 control versus UNESP-MM4 rapamycin treatment. The groups are described in Figure 7.

### 4.8. Proteomics

Cells were cultured in 75 cm^2^ bottles and after reaching 70% confluence, were divided in control group and rapamycin treatment (for 24 h) group. Then, cells were washed 3 times in cold DPBS buffer and after the last wash the supernatant was discarded and RIPA buffer (150 mM NaCl, 1% Triton X-100, 1% sodium deoxycholate, 0.1% SDS, 50 mM TRIS-HCl pH 7.5) containing protease inhibitors (50 mM TRIS-HCL pH 7.2, 0.8 mM EDTA, 1 µg/mL aprotinin, 1 µg/mL leupeptin, 35 µg/mL phenylmethylsulfonyl fluoride-PMSF), which was used for protein solubilization was added. Samples were sonicated (Sonifier Sound Encloresure Ultrasonics, Shanghai, China) for 3 cycles of 30 s followed by a 30 s pause. Then, the samples were centrifuged at 10,000× *g*, for 30 min, at 4 °C. Total protein concentration of samples was determined by a colorimetric method (Pierce™ 660nm Protein Assay, ThermoFisher Scientific, Pierce Biotechnology, Rockford, lL, USA) and nanospectrophotometry (NanoDrop 2000, Fisher Scientific Term™ Wilmington, DE, USA). 

For mass spectrometry samples were subjected to one-dimensional electrophoresis, however the system was turned off when the sample reached separation gel (12%), forming a single band. After the run, each band containing 50 µg of protein was cut out and the samples were digested according to [78] with modifications. The cut bands (fragments of ~ 1 mm) were incubated for 1 h at room temperature with 0.5 mL of a 50% methanol solution and 2.5% acetic acid in purified water (bleach solution 1). After that period, the solution was removed and another 0.5 mL of solution 1 (destain) was added. This step was performed three times, until the band completely discolored. The decolorization solution was removed by centrifugation and dehydrated with the addition of 200 µL of acetonitrile (100%), for 5 min. This dehydration step was performed two times. The sample was kept for 10 min, with the lid open, for the evaporation of the solvent. The reduction of the disulfide bridges was performed with 30 µL of 10 mM DTT. After that, samples were incubated for 30 min at room temperature. DTT was removed by centrifugation and alkylation was carried out with 50 mM iodoacetamide (30 µL) for 30 min at room temperature and protected from light. Again, samples were centrifuged to remove the iodoacetamide and gel fragments were washed with 100 µL of 100 mM ammonium bicarbonate, for 10 min, with centrifugation. Gel fragments were again dehydrated with 200 µL of acetonitrile (100%) by incubation for 5 min at room temperature. Acetonitrile was removed by centrifugation and the fragments were rehydrated with 200 µL of 100 mM ammonium bicarbonate, for 10 min. Bicarbonate was removed and the gel dehydrated with 200 µL of acetonitrile (100%), incubating for 5 min at room temperature. Acetonitrile was removed. This step was performed two times. The tube was kept open for solvent evaporation for 10 min. To each tube containing the sample was added 50 µL of trypsin at 20 ng/µL (Sequencing Grade Modified Trypsin, V5111-Promega), diluted in ammonium bicarbonate (50 mM). Excess trypsin was removed by centrifugation and 5 to 20 µL of ammonium bicarbonate (50 mM) were added until the fragments of the gel were covered. Overnight incubation was performed at 37 °C for approximately 16 h. Peptide extraction was carried out by adding 10 µL of extraction solution 1 (5% formic acid in ultrapure water), incubated for 10 min at room temperature. Samples were centrifuged and the supernatants stored. Further extraction was carried out with 5% formic acid in 50% acetonitrile, for 10 min. Fragments were centrifuged and the supernatant was removed and mixed with the first collected supernatant. The volume was reduced in an evaporator to 1 µL and the samples stored −20 °C.

For mass spectrometry, samples were thawed, diluted in 0.1% formic acid in the proportion of 0.7 µg of protein/µL, homogenized in a tube shaker and centrifuged at 1.100× *g* for 5 min. Then, 20 µL of the supernatant was deposited in specific tubes for analysis on the mass spectrometer (Clear glass 12 × 32 mm screw neck total recovery vial with lid, Waters Corporation, Milford, MA, USA). Mass spectrometry was performed on the LC ESI-QTOF equipment (Micromass Q-TOF PREMIER, Waters Corporation, Milford, MA, USA).

Protein analysis was performed according to Aragão et al. [79], an aliquot of 4.5 µL of protein resulting from peptide digestion was separated by column C18 (100 µm × 100 mm) RP nano UPLC (nanoAcquity, Waters Corporation, Milford, MA, USA) coupled to the LC ESI-QTOF mass spectrometer (Micromass Q-TOF PREMIER, Waters Corporation, Milford, MA, USA) with nanoelectrospray with a flow rate of 0.600 µL/min. The gradient was with 2–90% acetonitrile with 0.1% formic acid over 45 min. Voltage of the nanoelectrospray was 3.5 kV, voltage cone was 30 V at 100 uC. The device was operated in "top three" mode, in which an MS spectrum was acquired followed by MS/MS of the three most intense peaks detected. After fragmentation of MS/MS, the ion was placed on the exclusion list for 60 s and for the analysis of endogenous cleavage peptides, a real-time exclusion was used. Spectra were acquired using MassLynx v.4.1 software and the raw data files were converted to a peak list format (mgf) without adding the scans and searched in the UniprotSProt 10116 database (http://www.uniprot.org/) taxonomy *Canis lupus familiaris*, using the Mascot version 2.3.02 and Mascot Distiller MDRO version 2.4.0.0 tool (Matrix Science Inc, Boston, MA, USA), considering carbamidomethylation as fixed modifications, oxidation to methionine as variable modification, a trypsin cleavage, and 0.1 Da tolerance for fragment precursor ions.

After protein identification with ProteinPilot, data from the analysis with LC ESI-QTOF were inserted in the UniprotKB database (www.uniprot.org.br), to obtain the annotation of gene ontology using molecular function and biological process categories. Figures referring to gene ontology were obtained online at http://www.pantherdb.org, Panther version 10 [80].

For evaluation, cells were divided in control (UNESP-CM1, UNESP-CM60, UNESP-MM1 and UNESP-MM4) versus rapamycin treatment (UNESP-CM1, UNESP-CM60, UNESP-MM1 and UNESP-MM4), and in primary tumor control (UNESP-CM1 and UNESP-CM60) and metastasis control (UNESP-MM1 and UNESP-MM4) versus primary tumor rapamycin treatment (UNESP-CM1 and UNESP-CM60) and rapamycin treatment (UNESP-MM1 and UNESP-MM4). The experiment was performed in biological duplicates and in technical duplicates.

### 4.9. Statistical Analysis

#### 4.9.1. Immunofluorescence of Tumor Tissue

The results were represented by the mean and standard error of protein expression percentage.

#### 4.9.2. Cellular Metabolic Activity (MTT) and RT-qPCR

MTT results were analyzed by Tukey test and statistical differences were considered when *p* < 0.05. Results of the studied variables were evaluated for distribution using the Shapiro-Wilk normality test. The IC_50_ values were calculated using Graph Pad Prism 8.0 from a log ([drug]) versus normalized response curve fit.

Data obtained from the RT-qPCR were statistically analyzed using Mann-Whitney. Statistical differences were considered when *p* < 0.05 and results were presented as median and range values. 

#### 4.9.3. Proteomics

For proteomic results analysis data were normalized. The outliers were corrected when the sample contained ≥ 50% abundance of a protein; otherwise, they were reset to zero. Then, emPAI of each protein was divided by the sum of the emPAIs of that protein of all animals. These data were submitted to non-hierarchical cluster analysis (univariate and multivariate analysis) using the Metaboanalyst 3.0 software [81] (https://www.metaboanalyst.ca/MetaboAnalyst/upload/StatUploadView.xhtml). Groups were compared with the Student’s *t*-test (control versus treated) and ANOVA (all 4 groups). Moreover, principal component analysis (PCA) was used to describe the formation of the groups in a score matrix and the partial least squares-discriminant analysis (PLS-DA) which indicated the relevance of each protein in the division of the groups. PSL-DA was used to calculate the projection score of the variables (VIP score). The most relevant proteins and were considered when VIP score was at least α > 1 [82]. Venn diagram was created using the free software http://bioinformatics.psb.ugent.be/webtools/Venn/. Gene ontology results were obtained using Panther Classification System (http://pantherdb.org/geneListAnalysis.do) [83].

### 4.10. Data Availability

The original Mass Spectrometry dataset presented in the study are publicly available in Mendeley database here: http://dx.doi.org/10.17632/y69wn559ym.1 [84].

## 5. Conclusions

In conclusion, proteins found in primary tumor cells and their respective metastasis, treated or not with rapamycin, were related to tumor and cancer progression, invasion capacity, apoptosis and metastasis. Rapamycin treatment was able to inhibit cell growth and decrease cell viability in vitro. Although the treatment seemed to work in our cultured cells, the dose used to achieve this result was high when compared to the safely maximal dose used in dogs.

## Figures and Tables

**Figure 1 molecules-26-01213-f001:**
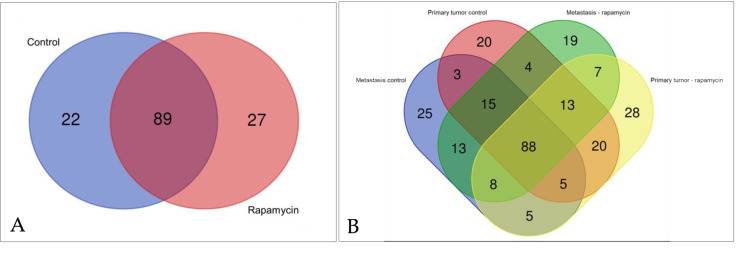
Venn diagram considering similar and different proteins found in all different groups studied. (**A**). represents all proteins found in control and rapamycin treated group. (**B**). shows proteins found in metastasis control, primary tumor control, metastasis rapamycin and primary tumor rapamycin groups.

**Figure 2 molecules-26-01213-f002:**
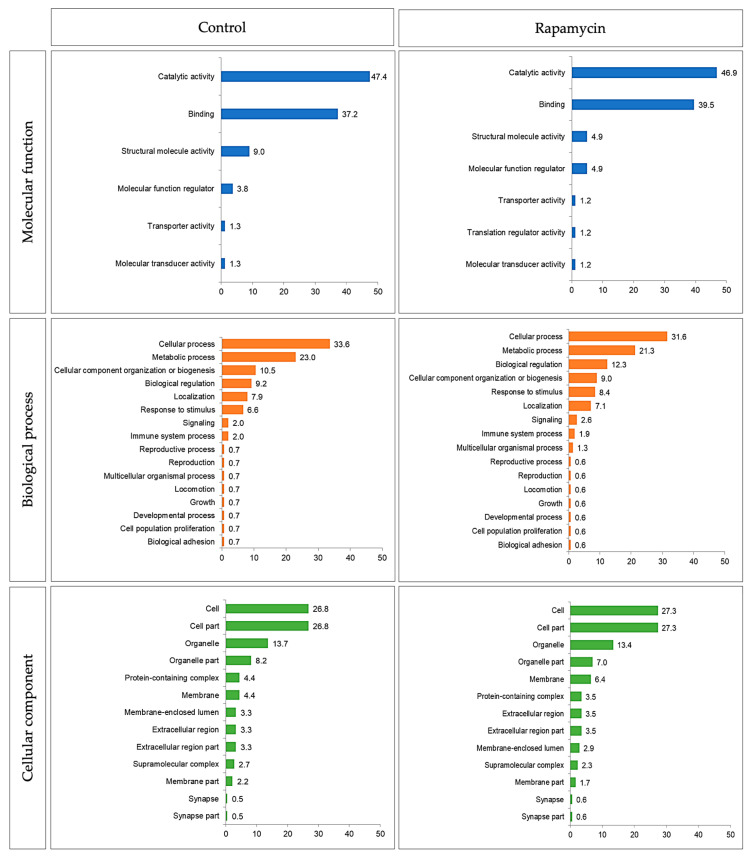
Gene ontology (molecular function, biological process, and cellular component) of control and rapamycin group.

**Figure 3 molecules-26-01213-f003:**
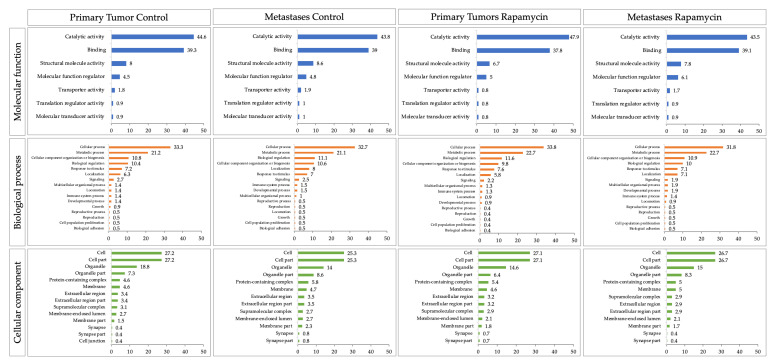
Gene ontology (molecular function, biological process, and cellular component) of primary tumor control, metastases control, primary tumor rapamycin and metastases rapamycin groups.

**Figure 4 molecules-26-01213-f004:**
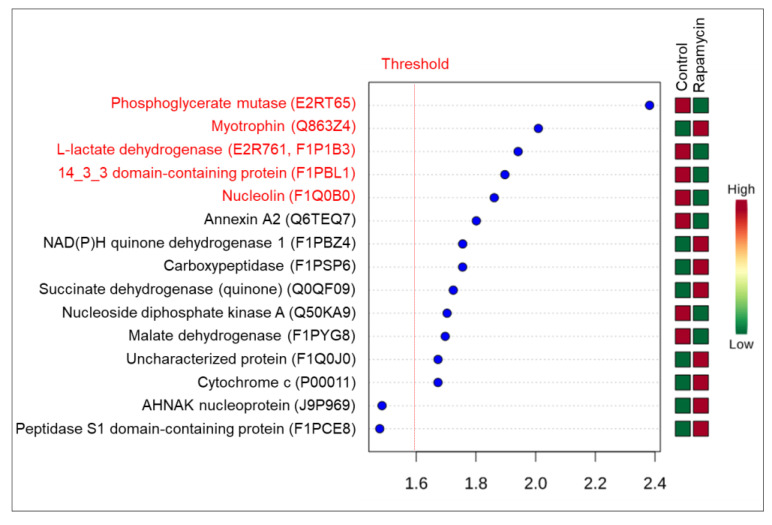
VIP score from PLS-DA of the control (non-treated versus treated with rapamycin). Colored boxes indicate protein abundance in each group (control and rapamycin treatment). Threshold considered for VIP score was α > 1.6. All proteins were observed in the T test and Volcano plot (FRD < 0.05). Proteins highlighted in red color were the main proteins in all analysis.

**Figure 5 molecules-26-01213-f005:**
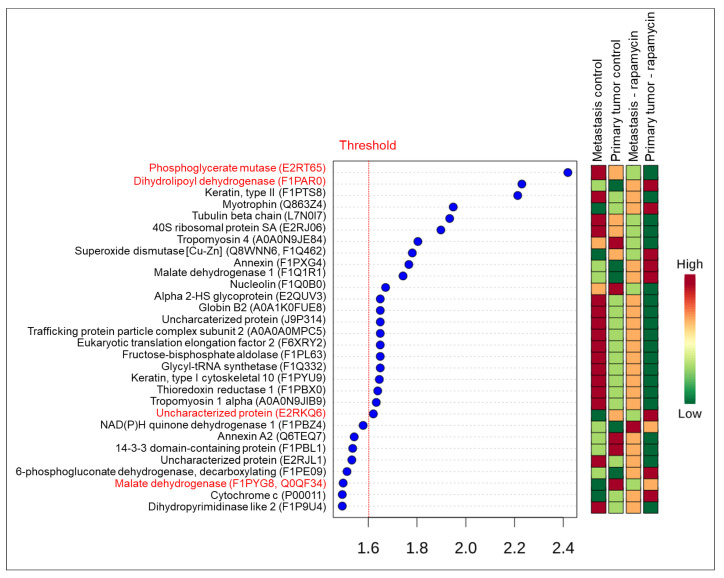
VIP score from PLS-DA. Colored boxes indicate the protein abundance in each group (metastasis control versus primary tumor control versus metastasis treated with rapamycin versus primary tumor treated with rapamycin). Threshold considered for VIP score was α > 1.6. Proteins in red color were also found in ANOVA one-way analysis (FRD < 0.05).

**Figure 6 molecules-26-01213-f006:**
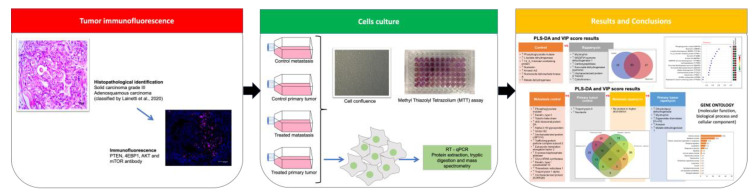
Experimental design.

**Figure 7 molecules-26-01213-f007:**
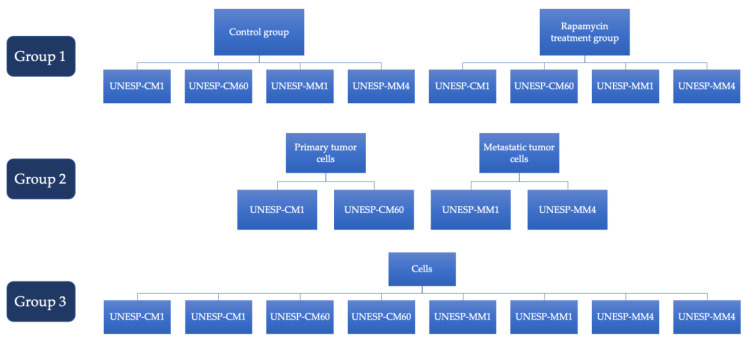
Experimental groups to perform RT-qPCR analysis.

## Data Availability

The data presented in this study are openly available in Mendeley database at doi:10.17632/y69wn559ym.1 [84].

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
