# Peer review of "Proteomics Approach of Rapamycin Anti-Tumoral Effect on Primary and Metastatic Canine Mammary Tumor Cells In Vitro"

_molecules, 2021, doi:10.3390/molecules26051213_

Round 1
Reviewer 1 Report
Dear Authors,
After lecture of the manuscript entitled ” Proteomics approach of rapamycin anti-tumoral effect on primary and metastatic canine mammary tumor cells in vitro” I recommend its major revision. Please find below comments, that should help in the manuscript correction:
- Line 19: bithces should be changed into bitches
- Line 46: sentence “The treatment of mammary tumors …”need to be reconsidered
- Line 154: “They were chosen because their importance in women breast cancer…” needs additional explanation. Comparison of the animal model to humans based on the expression level of 4 proteins is premature conclusion.
- Line 241: as basal like- as basal as
- Line 254: Nucelolin- Nucleolin
- Line 355, 386: Thermofisher (nomenclature should be unified in the whole manuscript)
- Line 356: Biotechnologia
- Line 392, 393: DNase
- Line 430: SDS-PAGE is not used for verification of the protein concentration (separation technique)
- Line 432: One-dimensional
- Line 471: nanoelectrospray
- Line 481: during proteomic analyses LC ESI-QTOF instrument was used, why there is an information regarding MALDI instrument?
- Line 485, 486: repetition
- Line 498: ..
- Considering the identification of protein content in cell homogenates, that is characterized with very large complexity and variability the number of analyzed samples and their repetition, is too small. Actually, the information about the number of technical and biological replicates during experiments is missing.
- The proteomic data should be presented as a results containing information: ion score, number of identified peptides, % sequence coverage…The table placed in supplementary materials is based only on Panther DB and UniProt KB, there are additional data that the readers might find by using protein ID. Mass spectrometry results should be attached to the manuscript.
- The total number of identified compounds in the cell homogenates is very poor, this might be the problem with MS analysis.
- Rapamycin mechanism should be studied more thoroughly, there are some studies that are deny its anti-tumoral activity [e.g. doi: 12688/f1000research.9207.1]
- Regrettably, the final conclusion “Some identified proteins could be used as markers…” is insufficient and needs to be discussed in more detailed.
Sincerely,
Reviewer 2 Report
The manuscript by Lainetti and colleagues entitled “ Proteomics approach of rapamycin anti-tumoral effect on primary and metastatic canine mammary tumor cells in vitro” describes a study on the identification of possible therapeutic targets for canine mammary tumors. In particular, the study includes the effect of Rapamycin on four canine tumor cell lines, expanding the knowledge of the biology of specific histo and immunotype of canine mammary tumors. Overall, the results of this work are of particular interest for future studies evaluating the therapeutic approaches for canine mammary tumors and several aspects are of particular interest also in human tumors. Moreover, a few specific points listed below need additional explanation.
Line 19. I suggest to reword the term “bitches” with female dogs all over the text
Line 46-49. This sentence is difficult to read and poorly written. Please reword it
Line 52: reference number 5 referred to human breast cancer. As this paper is focused on canine mammary tumor, I suggest to remove it and find, if it is the case, a more pertinent one.
Line 59: reference 12 is related to canine mammary tumors and not to osteosarcoma. Please revise the sentence
Line 321. Please, update the classification used according to the more recent one:“Surgical pathology of tumors of domestic animals, Volume 2: mammary tumors.”
Line 334: how the authors confirm the cross-reactivity of the primary antibodies with the dog tissue? Do you perform western blot? Please specify it as all the antibodies used are either monoclonal or polyclonal and the available online information about the reactivity does not include the canine specie.
Line 411-417: I will suggest to summarize the info with a graph
Line 500: I suggest to present the data as median and range values and box-plot as graph as, I assume, the data are not normally distributed,
Line 168-171: I’m not fully convinced about this statement, as there are several important differences between the classification system and the histotypes between the two species. Please, nuance the sentence.
Round 2
Reviewer 1 Report
Dear Authors,
Thank you for all your comments. I am pleased to recommend your manuscript for publication.